# BLIND CORESET SELECTION:
# EFFICIENT PRUNING FOR UNLABELED DATA

## ABSTRACT

Deep learning methods rely on massive data, resulting in substantial costs for storage, annotation, and model training. Coreset selection aims to select a representative subset of the data to train models with lower cost while ideally performing on par with the full data training. State-of-the-art coreset selection methods use carefully-designed criteria to quantify the importance of each data example using ground truth labels and dataset-specific training, then select examples whose scores lie in a certain range to construct a coreset. These methods work well in their respective settings, however, they cannot consider candidate data that are initially unlabeled. This limits the application of these methods, especially so considering that the majority of real-world data are unlabeled. To that end, this paper explores the problem of coreset selection for unlabeled data. We first motivate and formalize the problem of unlabeled coreset selection, which reduces annotation requirements to enable greater scale relative to label-based coreset selection. We then develop an unlabeled coreset selection method, *Blind Coreset Selection (BlindCS)*, that jointly considers overall data coverage on a distribution as well as the relative importance of each example based on redundancy. Notably, BlindCS does not use any model- or dataset-specific training, which increases coreset generalization and reduces computation relative to training-based coreset selection. We evaluate BlindCS on four datasets and confirm the advance over several state-of-the-art methods that use labels and training, leading to a strong baseline for future research in unlabeled coreset selection. Notably, the BlindCS coreset for ImageNet achieves a higher accuracy than previous label-based coresets at a 90% prune rate, while removing annotation requirements for 1.15 million images. We will make our code publicly available with the final paper.

## 1 INTRODUCTION

The computational cost to train a single state-of-the-art deep learning model in various fields doubles every 3.4 months in the deep learning era due to increasingly large models and datasets (Amodei et al., 2018; Zhao & Bilen, 2023). Since the introduction of AlexNet (Krizhevsky et al., 2012), groundbreaking models in computer vision like ViT and DALLE all rely on massive datasets for training (Dosovitskiy et al., 2021; Ramesh et al., 2022). However, there are substantial costs to collecting, storing, transmitting, and pre-processing such a vast amount of data. Furthermore, training models on vast datasets introduces yet another substantial cost for computation, sometimes hundreds of thousands of GPU hours to achieve satisfactory performance, which frustrates applications requiring repeat training over datasets such as hyparameter optimization (Maclaurin et al., 2015; Lorraine et al., 2020) and neural architecture search (Elsken et al., 2019; Li & Talwalkar, 2020).

Coreset selection deals with large data to mitigate the above issues for data-efficient deep learning. Specifically, coreset selection reduces the training set size by selecting a pruned subset that contains only valuable examples (the *core set*), such that models trained on the coreset achieve similar performance to those trained on the original, full dataset (Feldman et al., 2011). Several recent works provide various coreset selection methods using carefully-designed criteria, including median class values (Xia et al., 2023), diverse coverage of importance scores (Zheng et al., 2023), and gradient dynamics during training (Zhang et al., 2024), which achieves 53.91% accuracy on ImageNet with only 10% training data.

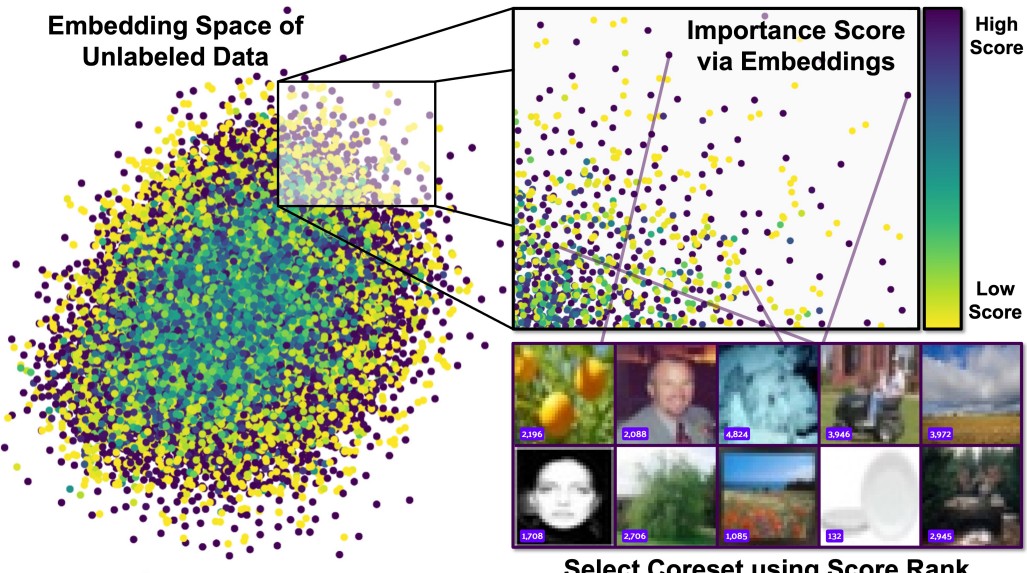

Figure 1: **Blind Coreset Selection Overview**. To select coresets from unlabeled data, we first use off-the-shelf models to generate a dataset embedding space (e.g., a 2-D slice of CLIP on CIFAR100, left). Using the embeddings, we calculate an importance score that rewards examples individually covering large portions of the embedding space while penalizing immediate neighbors to remove redundancy. Finally, we output a coreset of examples for any given prune rate using the score rank. Embeddings and data visualizations generated using the FiftyOne Library (Moore & Corso, 2020).

State-of-the-art coreset selection methods have demonstrated impressive results in experiment settings. However, the current SOTA methods assume the full dataset is labeled and available for training prior to coreset selection. Regarding labels, it is important to acknowledge that the majority of real-world data are, in fact, unlabeled, preventing coreset consideration for label-based methods. Furthermore, labeling massive amounts of image data just to consider selection is cost prohibitive, with annotation taking anywhere between 7 s per bounding box to 1.5 hours for full semantic segmentation (Jain & Grauman, 2013; Cordts et al., 2016). Some innovative coreset selection methods use self-supervised learning in place of label-based training (Sorscher et al., 2022); however, this approach will still have substantial time and computation costs to select coresets at scale. Furthermore, coupling coreset selection with training on a single model architecture decreases generalization.

To that end, this paper addresses the problem of coreset selection without labels or training using a novel approach. First, we formulate the problem of unlabeled coreset selection, which reduces data- *and* label-based costs by generating coresets from unlabeled data. After coreset selection from the larger dataset, labels are *only* used by the actual model to train on the pruned dataset. Notably, if coreset selections are for self-supervised training, no labels are used. Second, we use the unlabeled coreset selection formulation to develop Blind Coreset Selection (BlindCS), a method which *also* reduces computation costs by selecting coresets without training on the candidate dataset. Instead, BlindCS uses off-the-shelf models to generate a candidate selection embedding space, which is then iteratively sampled and scored to estimate the value of each example's value based on coverage of the embedding space and redundancy within the coreset (see Figure 1).

Our contributions are as follows:

1. We motivate and formalize the problem of unlabeled coreset selection, which substantially reduces data- *and* label-based costs for efficient deep learning at scale.

2. We develop our Blind Coreset Selection method (BlindCS), which is computationally efficient and uses novel estimates of dataset distribution coverage and redundancy to select coresets from larger, unlabeled datasets, enabling broader application.

3. We evaluate BlindCS against state-of-the-art label- and training-based coreset selection methods with eight baselines on four different datasets spanning three orders of magnitude

Table 1: Comparison of data and procedural requirements across coreset selection methods.

| Methods | Selects Coreset Data | | |
| --- | --- | --- | --- |
| | *without* Training on Data | *without* Ground Truth Labels | *without* Prune Specific Tuning |
| **Blind Coreset Selection (ours)**, Random | **Yes** | **Yes** | **Yes** |
| Self-Supervised Selection NeurIPS 2022 | No | **Yes** | **Yes** |
| Moderate ICLR 2023, Dyn-Unc CVPR WS '24 | No | No | **Yes** |
| TDDS CVPR 2024, Coverage ICLR 2023 | No | No | No |

for scale. Results demonstrate that our method performs best in multiple cases and overall outperforms all label-based methods save one, while reducing label and computation costs.

From these results, BlindCS sets a new state-of-the-art for coreset selection work.

## 2 RELATED WORK

**Dataset Distillation** is similar to coreset selection in that it comprises many innovative methods for data-efficient deep learning. On a functional level, the objectives of many coreset methods also apply to dataset distillation, however, as opposed to selecting a subset of *existing* data for a coreset, dataset distillation aims to generate a much smaller dataset with *synthetic* examples that yield the same performance as the larger initial dataset (Yu et al., 2024). Notable dataset distillation methods generate synthetic examples relative to the initial dataset by matching gradients (Zhao et al., 2021), differentiable Siamese augmentation for better synthesis (Zhao & Bilen, 2021), aligning features (Wang et al., 2022), multi-step parameter matching (Cazenavette et al., 2022), and embedding space distribution matching (Zhao & Bilen, 2023). These dataset distillation methods are remarkable for their creation of small but effective synthetic training datasets. On the other hand, our current work focuses on evaluating and selecting coresets from existing real-world data.

**Active Learning** is another active research area with many contributions to data-efficient deep learning. The goal of active learning is to enable learning algorithms to perform better with less training by letting them choose their own data (Settles, 2012), which is especially useful in cases where large portions of data are unlabeled and manual labeling is expensive (Bernard et al., 2018). In fact, active learning encompasses the particularly hard problem of starting selection with no initial labeled examples, i.e., the cold start problem (McCallum & Nigam, 1998). Notably, some recent active learning methods focus on the importance of coverage diversity in data selection (Ash et al., 2020; Citovsky et al., 2021). However, these methods actively train and select data on an increasing set for a specific model, which is not conducive for *model-agnostic*, *one-shot* coreset selection.

**Coreset Selection** prunes datasets down to a smaller, valuable *core set* to reduce costs and enable more data-efficient deep learning. A basic solution to find the optimal coreset is to search through and train on every subset to find the best corresponding model performance. However, this simple approach is NP-hard, which has led to the development of many innovative coreset selection methods. Early coreset methods generally expect a consistent data distribution to the original dataset (Feldman et al., 2011; Bachem et al., 2015), e.g., Welling (2009) greedily adds one sample at a time to match embedding space centers. Other coreset methods can be broadly categorized as selecting by optimization (Wei et al., 2015; Yang et al., 2023), coverage or diversity (Sener & Savarese, 2018; Zheng et al., 2023), and importance criteria (Toneva et al., 2019; Tan et al., 2023). Recent coreset innovations address ongoing challenges such as application on a wide range of dataset sizes (Xia et al., 2023), making selections on data with label errors (Park et al., 2023), and fully utilizing training dynamics (Zhang et al., 2024).

Our current work is inspired by the success of this previous coreset selection work. However, a drawback for current state-of-the-art coreset selection methods is requiring labels and/or training on the larger initial dataset (see Table 1). Thus, in this paper, we focus on extending coreset selection to unlabeled data without any requirements for dataset- or architecture-specific training. This broadens general applicability to new data and models while reducing costs associated with annotating data with ground truth labels, sensitivity to label errors, and extensive computation at scale.

## 3 PRELIMINARIES

We define the problem of labeled coreset selection for data-efficient deep learning. Formally, we are given a labeled dataset $\mathbb{S}^{\text{L}} = \{(\mathbf{x}_i, y_i)\}_{i=1}^N$ with $N$ examples drawn i.i.d. from an underlying distribution $P$, where $\mathbf{x}_i$ are the data and $y_i$ is the ground truth label for each example. The goal is to select a subset of $\mathbb{S}^{\text{L}}$ to reduce future storage and training consumption while closely maintaining performance of full dataset training. We denote this *coreset* as $\mathbb{S}^{\text{C}} = \{(\mathbf{x}_i, y_i)\}_{i=1}^n \subset \mathbb{S}^{\text{L}}$, which has $n$ examples and a *prune rate* of $\frac{(1-n)}{N}$. We formulate coreset selection as (Sener & Savarese, 2018):

$$\underset{\mathbb{S}^{\text{C}} \subset \mathbb{S}^{\text{L}} \mid \frac{1-n}{N} \geq p}{\arg\min} \; \mathbb{E}_{\mathbf{x}, y \sim P}[l(\mathbf{x}, y; f_{(\mathbb{S}^{\text{c}})})], \tag{1}$$

where $p$ is a prune rate set *before* training, $l$ is the loss function, and $f_{(\mathbb{S}^{\text{c}})}$ is a model trained on $\mathbb{S}^{\text{C}}$. Notably, many SOTA methods select $\mathbb{S}^{\text{C}}$ by assigning an importance score to each example (e.g., Zhang et al. (2024)). For later use, we denote the importance score as $\boldsymbol{s} \in \mathbb{R}^N$.

## 4 UNLABELED CORESET SELECTION

We define the problem of unlabeled coreset selection for data- and *label*-efficient deep learning. Formally, given an unlabeled dataset $\mathbb{S} = \{(\mathbf{x}_i)\}_{i=1}^N$, the goal is to select $\mathbb{S}^{\text{C}} \subset \mathbb{S}$ without using *any* ground-truth label $y_i$. The motivation for this change is that it is preventative to label an entire massive dataset when much of the data will be pruned. We formulate unlabeled coreset selection by replacing $\mathbb{S}^{\text{C}} \subset \mathbb{S}^{\text{L}}$ with $\mathbb{S}^{\text{C}} \subset \mathbb{S}$ in Equation (1). Notably, after selecting $\mathbb{S}^{\text{C}}$, we add $n$ labels to the coreset as $\mathbb{S}^{\text{C}} = \{(\mathbf{x}_i, y_i)\}_{i=1}^n$ *only* to train the pruned model $f_{(\mathbb{S}^{\text{c}})}$.

Along with the aforementioned benefits of coreset selection, unlabeled coreset selection uniquely increases scale and reduces labeling costs. First, while we can use any $\mathbf{x}_i$ from a labeled dataset $\mathbb{S}^{\text{L}}$, we can also extensibly sample and consider more examples $\mathbf{x}$ from the underlying distribution $P$ without any annotation or labeling requirements. This extension enables us to source coresets from a much larger initial dataset. In effect, unlabeled coreset selection extends dataset pruning to the majority of unlabeled, real-world data. Second, we only label the $n$ coreset examples after they are selected for pruned model training, so there is a $N - n$ reduction in labeling costs relative to label-based coreset selection. As one specific example, using unlabeled coreset selection at a 90% prune rate on ImageNet removes label requirements for 1.15 million images.

## 5 METHODOLOGY

Using the unlabeled coreset selection formulation, we develop a new method of "Blind" Coreset Selection (BlindCS). In place of label- or training-based selection, BlindCS alternatively uses an off-the-shelf model embedding space representation of the initial dataset (Section 5.1). BlindCS then samples the embedding space to determine which examples provide valuable coverage (Section 5.2). Subsequently, BlindCS determines which examples in proximity to those providing coverage are redundant (Section 5.3). Finally, BlindCS uses the coverage and redundancy metrics to iteratively sample and score each candidate training example to determine final coreset selections (Section 5.4).

### 5.1 FOUNDATIONAL EMBEDDING REPRESENTATION

BlindCS uses an embedding space representation of unlabeled dataset $\mathbb{S}$. To generate embeddings in this work, we use an off-the-shelf deep learning model denoted as $f(\cdot) = g(h(\cdot))$, where $h$ is the model component mapping input data to hidden representations at the penultimate layer and $g$ maps the embedding space to a previously learned output $f$. We use $h(\mathbf{x}_i) \in \mathbb{R}^M$ to generate an $M$-dimension *embedding space* for input data $\mathbb{S} = \{(\mathbf{x}_i)\}_{i=1}^N$ denoted as

$$\boldsymbol{Z} = [h(\mathbf{x}_1), \cdots, h(\mathbf{x}_N)] \in \mathbb{R}^{N \times M}. \tag{2}$$

Notably, Equation (2) lets us to use the previously learned hidden representation of $h$ as an alternative to label- or training-based coreset selection. Instead, we quantify the importance of each example in terms of relative coverage (Section 5.2) and redundancy (Section 5.3) in feature-based embedding space $\boldsymbol{Z}$ as a representation of the underlying data distribution $\mathbf{x}, y \sim P$ in Equation (1).

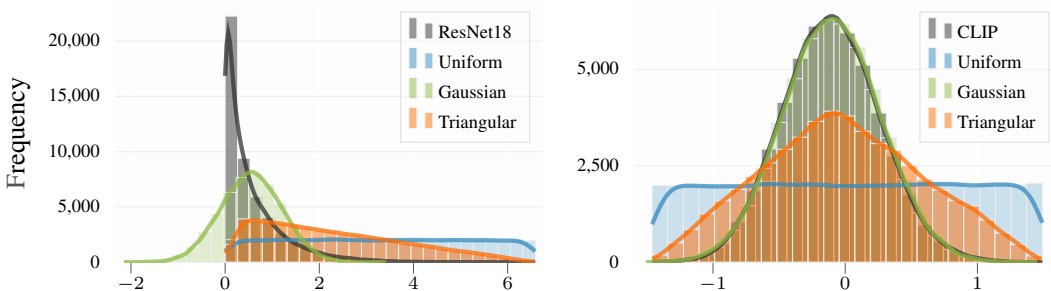

Figure 2: Comparison of real embedding data (gray) and **sampling techniques**. ResNet18 (left) and CLIP (right) are the first dimension embeddings for 50,000 CIFAR100 train set examples, while each corresponding distribution type is sampled 50,000 times. Relative to uniform or Gaussian, our Triangular distribution uniquely achieves all objectives of: providing ample coverage for densely populated regions of the embedding space, covering outliers, and not oversampling empty space.

**Remarks on $Z$**: For experiments in Section 6, we generate all model embeddings in advance using off-the-shelf weights for a ResNet18 (He et al., 2016) and CLIP ViT-L-14 model (Radford et al., 2021), which we concatenate as $h(\mathbf{x}_i) = \begin{bmatrix} h^{\text{RN18}}(\mathbf{x}_i) \\ h^{\text{CLIP}}(\mathbf{x}_i) \end{bmatrix} \in \mathbb{R}^{1,280}$. Notably, relative to coreset methods using full dataset training for 60-200 epochs, embedding space generation for BlindCS takes less time than one epoch given that we use only one forward pass per sample, a subcomponent of the overall model architecture ($h$), and no training-based back propagation or metric tracking.

## 5.2 COVERAGE OF THE EMBEDDING SPACE

Our first objective for coreset selection is to select examples that maximize coverage of embedding space $Z$. To quantify coverage, we develop a Monte Carlo-inspired sampling technique (Metropolis & Ulam, 1949), which estimates the relative contribution of each candidate training example $\mathbf{x}_i \in \mathbb{S}$ in covering a carefully designed distribution over the embedding space.

We assume a Triangular distribution over each embedding space dimension $j \in \{1, \cdots, M\}$ using

$$\mathbf{s}_j \sim p(\mathbf{x}, j) := \begin{cases} \frac{2(\mathbf{x} - z^{\min}{}_j)}{(z^{\max}{}_j - z^{\min}{}_j)(z^{\text{med}}{}_j - z^{\min}{}_j)} & \text{for } z^{\min}{}_j \leq \mathbf{x} < z^{\text{med}}{}_j \\ \frac{2(z^{\max}{}_j - \mathbf{x})}{(z^{\max}{}_j - z^{\min}{}_j)(z^{\max}{}_j - z^{\text{med}}{}_j)} & \text{for } z^{\text{med}}{}_j \leq \mathbf{x} \leq z^{\max}{}_j \end{cases},$$
$$\mathbf{s} := [\mathbf{s}_1, \cdots, \mathbf{s}_M]^{\mathsf{T}} \in \mathbb{R}^M, \tag{3}$$

where $\mathbf{s}$ is a full random sample of $Z$, $\boldsymbol{z}^{\min} = \{\min(\boldsymbol{Z}_{:,j})\}_{j=1}^M \in \mathbb{R}^M$ is the minimum $Z$ value for each embedding dimension, and $\boldsymbol{z}^{\text{med}}, \boldsymbol{z}^{\max} \in \mathbb{R}^M$ are the corresponding median and maximum $Z$ values. In practice, our Triangular distribution robustly covers both exponential- (ResNet) and Gaussian-shaped (CLIP) embedding distributions, naturally balancing between common and fringe embeddings as shown in Figure 2.

We increase sample efficiency over $\boldsymbol{Z} \in \mathbb{R}^{N \times M}$ by reducing its dimensionality to $\mathbb{R}^{N \times m}$ using

$$\boldsymbol{D} := [\mathbf{1}_{d_1}, \cdots, \mathbf{1}_{d_m}] \in \mathbb{R}^{M \times m},$$
$$\hat{\boldsymbol{Z}} := \boldsymbol{Z}\boldsymbol{D} \in \mathbb{R}^{N \times m}, \tag{4}$$

where $\boldsymbol{D}$ linearly maps $\boldsymbol{Z}$ to $m$ reduced embedding dimensions, $\mathbf{d} = [d_1, \cdots, d_m]^{\mathsf{T}} \in \mathbb{N}^m$ is a set of random indices chosen without replacement from $\{1, \cdots, M\}$, and $\mathbf{1}_i$ is a one-hot vector with $i$-th element equal to 1. In plain words, we use $\boldsymbol{D}$ to randomly select a subset of $m \leq M$ indices to represent $\boldsymbol{Z}$ in a lower dimensional subspace $\hat{\boldsymbol{Z}}$. In addition to $\boldsymbol{Z}$, we similarly reduce the dimension of random sampling $\mathbf{s} \in \mathbb{R}^M$ in Equation (3) using Equation (4) to find $\hat{\mathbf{s}} := \mathbf{s}\boldsymbol{D} \in \mathbb{R}^m$.

We quantify coverage for each random sample $\hat{\mathbf{s}}$ by finding the closest *existing* dataset example

$$\arg\min_i ||\hat{\mathbf{s}} - \hat{\boldsymbol{Z}}_i||_1, \tag{5}$$

where we denote $k$ as the solution to $i$ in Equation (5) and $\hat{\boldsymbol{Z}}_k$ is the dataset example closest to $\hat{\mathbf{s}}$. Finally, we quantify our importance score for coverage ($\boldsymbol{s}^{\mathrm{C}}$) as

$$s_i^{\mathrm{C}} := \begin{cases} 1 & \text{for } i = k \\ 0 & \text{otherwise} \end{cases}, \tag{6}$$

$$\boldsymbol{s}^{\mathrm{C}} := [s_1^{\mathrm{C}}, \cdots, s_N^{\mathrm{C}}] \in \mathbb{R}^N,$$

where $\boldsymbol{s}^{\mathrm{C}}$ adds to the estimated embedding coverage value for dataset example $k$. We repeat our process of randomly sampling $\hat{\mathbf{s}}$ and subsequently adding coverage for the closest examples across many iterations, which extends our estimated coverage score across all examples in $\mathbb{S}$. Unlike random sampling, our coverage score rewards hard examples that individually occupy large, unique, low-density areas of the overall embedding space (see Figure 1), which improves coreset selection.

**Remarks on** $m$: For experiments in Section 6, we choose $m = 2$ (s.t. $D \in \mathbb{R}^{M \times 2}$) random embedding dimensions per sample $\hat{\mathbf{s}}$, which increases computational efficiency on large datasets while enabling $\binom{M}{2} \approx \frac{M^2}{2}$ unique 2-D embedding space slices of $\boldsymbol{Z}$ over numerous sampling iterations.

## 5.3 REMOVING EMBEDDING SPACE REDUNDANCY

To avoid redundant coreset selection in the embedding space, we develop a corresponding redundancy estimate that operates subsequently to each coverage solution $k$ in Equation (5). Specifically, for each coverage example $\hat{\boldsymbol{Z}}_k$, we quantify redundancy for the set of $\mathbb{K} \in \mathbb{N}^\alpha$ nearest neighbors as

$$\boldsymbol{v}^{\mathrm{R}} := \begin{cases} \left(||\hat{\boldsymbol{Z}}_k - \hat{\boldsymbol{Z}}_i||_1\right)^{-\beta} & \text{for } i \in \mathbb{K} \\ 0 & \text{otherwise} \end{cases}, \tag{7}$$

where exponential $\beta$ determines how quickly the penalty changes between neighbors with varying distances to $\hat{\boldsymbol{Z}}_k$ of $||\hat{\boldsymbol{Z}}_k - \hat{\boldsymbol{Z}}_i||_1$. Using $\boldsymbol{v}^{\mathrm{R}} \in \mathbb{R}^N$, we define our redundancy score as

$$\boldsymbol{s}^{\mathrm{R}} := \frac{\boldsymbol{v}^{\mathrm{R}}}{||\boldsymbol{v}^{\mathrm{R}}||_1}, \tag{8}$$

where $||\boldsymbol{v}^{\mathrm{R}}||_1 \in \mathbb{R}$ normalizes $\boldsymbol{s}^{\mathrm{R}} \in \mathbb{R}^N$ so that the coverage and redundancy scores for each sample iteration are balanced as $||\boldsymbol{s}^{\mathrm{R}}||_1 = ||\boldsymbol{s}^{\mathrm{C}}||_1 = 1$.

**Remarks on** $\alpha, \beta$: For experiments in Section 6, we choose $\alpha = 1{,}000$ to limit computation of Equation (7) on large datasets while still reaching many examples per iteration, and we choose $\beta = 4$ to ensure that primarily the closest neighbors to each $\hat{\boldsymbol{Z}}_k$ are substantially estimated as redundant.

## 5.4 PRUNING PROCEDURE

Using the embedding sampling process for $\hat{\mathbf{s}}$ in Equation (5) and subsequent coverage $\boldsymbol{s}^{\mathrm{C}}$ and $\boldsymbol{s}^{\mathrm{R}}$ scores, we define our final importance score $\boldsymbol{s} \in \mathbb{R}^N$ as

$$\boldsymbol{s} := \sum_{t=1}^T \boldsymbol{s}_t^{\mathrm{C}}(\hat{\mathbf{s}}_t) - \boldsymbol{s}_t^{\mathrm{R}}(k_t), \tag{9}$$

where $\hat{\mathbf{s}}_t$ is the random embedding space sample $\hat{\mathbf{s}}$ at iteration $t$ with corresponding coverage score $\boldsymbol{s}_t^{\mathrm{C}}(\hat{\mathbf{s}}_t)$, $k_t$ is the example solution in Equation (5) at iteration $t$ with corresponding redundancy score $\boldsymbol{s}_t^{\mathrm{R}}(k_t)$, and $T$ is the overall number of sample and score iterations. Notably, each iteration $t$ is independent, which enables us parallelize our importance score for accelerated computation.

Finally, after finding $\boldsymbol{s}$ as our importance score to rank all examples in unlabeled dataset $\mathbb{S}$, we select the $n$ examples with highest scores as our pruned coreset for model training.

For experiments in Section 6, we also use $\boldsymbol{s}$ to weight the loss and gradient for model training using

$$\boldsymbol{w} = \frac{\boldsymbol{s} + \min(\boldsymbol{s})}{\max(\boldsymbol{s}) - \min(\boldsymbol{s})}, \tag{10}$$

where $\boldsymbol{w} = [w_1, \cdots, w_N]^\mathsf{T} \in \mathbb{R}^N$, $w_i \in [0, 1]$, and the loss is scaled each batch by the mean $w_i$ score corresponding to the specific training examples in that batch. Basically, we already assign a value to each example for coreset selection and want to influence model training accordingly.

Table 2: Comparison of **full training and coreset size** across all datasets. Prune rate is the % of training data removed. BlindCS uses constant parameter settings across all datasets and prune rates and, relative to label-based selection methods, removes labeling requirements from the full dataset.

| Dataset | Scale | Number of Classes | Full Dataset Training Size | Coreset Size at Various Prune Rates | | | | |
|---------|-------|-------------------|----------------------------|------|------|------|------|------|
| | | | | 30% | 50% | 70% | 80% | 90% |
| ImageNet | Large | 1,000 | 1,281,167 | 896,817 | 640,584 | 384,350 | 256,233 | 128,117 |
| CIFAR100 | Medium | 100 | 50,000 | 35,000 | 25,000 | 15,000 | 10,000 | 5,000 |
| CIFAR10 | Medium | 10 | 50,000 | 35,000 | 25,000 | 15,000 | 10,000 | 5,000 |
| EuroSAT 80 | Medium | 10 | 21,600 | 15,120 | 10,800 | 6,480 | 4,320 | 2,160 |
| EuroSAT 40 | Small | 10 | 10,800 | 7,560 | 5,400 | 3,240 | 2,160 | 1,080 |
| EuroSAT 20 | Small | 10 | 5,400 | 3,780 | 2,700 | 1,620 | 1,080 | 540 |
| EuroSAT 10 | Small | 10 | 2,700 | 1,890 | 1,350 | 810 | 540 | 270 |

## 6 EVALUATION

### 6.1 EXPERIMENTAL SETUP

**Datasets**. We evaluate the effectiveness of Blind Coreset Selection (BlindCS) on four image classification datasets: CIFAR10 (Krizhevsky, 2009), CIFAR100, ImageNet (Deng et al., 2009), and EuroSAT (Helber et al., 2019). We compare the full training and coreset size across each dataset in Table 2. Notably, full dataset sizes span from 1.3 M to 2,700 examples and coreset sizes span from 896,817 to 270 examples (three orders of magnitude). EuroSAT has no explicit training set, so we create "four" datasets using 80/20, 40/60, 20/80, and 10/90 training/validation splits to experiment with dataset scale in the same distribution of satellite images.

**Network Training**. We use two different network models and training regimes to evaluate coresets. For CIFAR10, CIFAR100, and EuroSAT, we train a ResNet18 model on selected coresets for 200 epochs with a batch size of 128. For ImageNet, we alternatively train a ResNet32 model for 60 epochs with a batch size of 256. Following the protocol of Zhang et al. (2024), we use an SGD optimizer with momentum 0.9, weight decay 0.0005, and a learning rate of 0.1 that decays with the cosine annealing scheduler via PyTorch (Paszke et al., 2019). After model training, we use the model's validation accuracy to quantitatively evaluate coreset selection performance.

**BlindCS & Baselines**. We implement BlindCS using the Section 5 formulation with constant parameter settings across all datasets and prune rates. We compare BlindCS against the current state-of-the-art using eight methods. BlindCS is the only method that does not use ground truth labels and dataset training aside from **Random**, which selects examples with uniform random sampling. **Entropy** selects examples with high entropy of predicted probabilities at the end of training (Coleman et al., 2020). **Forgetting** selects examples that change to being misclassified after correct classification the most times during training (Toneva et al., 2019). **EL2N** selects examples with high gradient magnitude using the L2 norm of error vectors (Paul et al., 2021). **AUM** selects examples with high area under the margin, i.e., the probability gap between between the target class and the next largest class across all epochs (Pleiss et al., 2020). **Moderate** selects examples closest to the median class value in the full dataset trained model embedding space (Xia et al., 2023). **Dyn-Unc** selects examples with high target class probability variance during training (He et al., 2024). Finally, **TDDS** selects examples with high projected gradient variance across many epochs (Zhang et al., 2024).

### 6.2 CORESET PERFORMANCE COMPARISON

We provide coreset selection results for CIFAR10 and CIFAR100 in Table 3, which demonstrates coreset selection on two medium-sized datasets. Relative to CIFAR10, CIFAR100 is more challenging with an order of magnitude more classes. Across both datasets, BlindCS achieves the best performance over all label- and training-based methods at all prune rates, with the exception of TDDS, which is a label- and training-based method. Notably, BlindCS and TDDS are the only methods outperforming Random, with the largest relative performance gaps between methods occurring at high prune rates.

Table 3: Comparison of Unlabeled and Labeled coreset selection methods on **CIFAR10** and **CI-FAR100**. Full dataset training on the ResNet18 model achieves 95.23% (CIFAR10) and 78.21% (CIFAR100) accuracy. Prune rate is the % of training data removed. "Rel. Rand." is Mean accuracy across all prune rates on both datasets relative to Random. BlindCS and TDDS prune selections outperform all other methods and Random on both datasets. A results plot is provided in the Appendix.

| | CIFAR10 | | | | | CIFAR100 | | | | | Mean Rel. Rand. |
|---|---|---|---|---|---|---|---|---|---|---|---|
| Prune Rate | 30% | 50% | 70% | 80% | 90% | 30% | 50% | 70% | 80% | 90% | |
| **Unlabeled Coreset Selection without Training** | | | | | | | | | | | |
| **BlindCS** | **94.58** | **93.46** | **90.97** | **89.06** | **84.18** | **76.04** | **72.87** | **65.92** | **61.92** | **52.11** | **78.11** |
| | ±0.09 | ±0.16 | ±0.17 | ±0.33 | ±0.21 | ±0.15 | ±0.18 | ±0.15 | ±0.39 | ±0.66 | **+1.34** |
| Random | **94.58** | 93.38 | 90.61 | 88.87 | 83.77 | 75.53 | 71.95 | 64.59 | 57.79 | 46.68 | 76.78 |
| | ±0.04 | ±0.17 | ±0.44 | ±0.47 | ±0.26 | ±0.04 | ±0.16 | ±0.32 | ±0.24 | ±1.07 | +0.00 |
| **Labeled Coreset Selection with Training-based Pruning** | | | | | | | | | | | |
| TDDS | **95.47** | **95.21** | **93.03** | **91.30** | **85.46** | **77.56** | **74.04** | **67.78** | **63.01** | **54.51** | **79.74** |
| CVPR 2024 | ±0.06 | ±0.04 | ±0.25 | ±0.21 | ±0.21 | ±0.06 | ±0.34 | ±0.44 | ±0.12 | ±0.22 | **+2.96** |
| Moderate | 93.96 | 92.34 | 89.71 | 87.75 | 83.61 | 74.60 | 70.29 | 62.81 | 56.52 | 41.82 | 75.34 |
| ICLR 2023 | ±0.06 | ±0.09 | ±0.14 | ±0.27 | ±0.24 | ±0.10 | ±0.31 | ±0.08 | ±0.37 | ±1.12 | -1.43 |
| Entropy | 94.45 | 91.90 | 86.24 | 83.49 | 72.06 | 72.39 | 64.44 | 50.73 | 42.86 | 29.56 | 68.81 |
| ICLR 2020 | ±0.07 | ±0.16 | ±0.26 | ±0.21 | ±0.81 | ±0.20 | ±0.36 | ±0.86 | ±0.25 | ±0.54 | -7.96 |
| Forgetting | 95.45 | 95.05 | 89.14 | 76.18 | 45.87 | 77.38 | 70.76 | 49.92 | 38.42 | 25.82 | 66.40 |
| ICLR 2019 | ±0.24 | ±0.05 | ±2.04 | ±3.18 | ±1.87 | ±0.09 | ±0.40 | ±0.28 | ±1.13 | ±0.52 | -10.38 |
| Dyn-Unc | 95.08 | 94.03 | 89.40 | 79.76 | 37.12 | 73.36 | 65.90 | 50.16 | 39.19 | 15.20 | 63.92 |
| CVPR WS '24 | ±0.02 | ±0.14 | ±0.13 | ±1.09 | ±1.12 | ±0.10 | ±0.25 | ±0.47 | ±0.27 | ±0.41 | -12.86 |
| AUM | 95.44 | 95.19 | 91.19 | 69.60 | 34.74 | 77.35 | 68.17 | 31.69 | 18.43 | 9.29 | 59.11 |
| NeurIPS 2020 | ±0.09 | ±0.09 | ±0.63 | ±3.11 | ±0.11 | ±0.18 | ±0.52 | ±0.34 | ±0.47 | ±0.27 | -17.67 |
| EL2N | 95.43 | 95.06 | 86.69 | 68.64 | 31.89 | 76.89 | 67.57 | 36.45 | 17.31 | 9.10 | 58.50 |
| NeurIPS 2021 | ±0.10 | ±0.04 | ±1.71 | ±3.70 | ±1.51 | ±0.31 | ±0.15 | ±1.36 | ±0.33 | ±0.69 | -18.27 |

We provide coreset selection results for ImageNet in Table 4, which demonstrates coreset selection at a large scale. Overall, BlindCS and TDDS coreset selections outperform all other methods. Notably, BlindCS selects the best performing coreset at the 90% prune rate without using any labels, which removes label requirements for 1.15 million images.

We plot coreset selection results for all EuroSAT dataset splits in Figure 3, which demonstrates coreset selection for the three leading methods at a much smaller scale. Except for the 90% prune rate on small datasets, BlindCS cuts much of the performance gap between unlabeled Random selection and label- and training-based TDDS. For 90% prune rates, BlindCS outperforms TDDS on EuroSAT 40 but has a lower accuracy than TDDS and Random on EuroSAT 20 and EuroSAT 10, where the pruned coresets only have 540 and 270 training examples. Notably, unlike TDDS, BlindCS is currently using constant parameter settings across all prune rates. On the other hand, BlindCS small dataset performance improves with alternative settings (e.g., reducing the number of nearest neighbors for redundancy in Equation (7)), which we will address in future work.

## 6.3 ABLATION STUDY

We provide BlindCS ablative results in Table 5. When using a single model to generate our embedding space ($Z$), ResNet18 outperforms CLIP, but neither perform as well as the standard concatenated setting. Gaussian sampling ($s$) outperforms uniform but does not match Triangular performance. However, given the narrow performance gap between Triangular and Gaussian sampling, we postulate that exploring additional sampling strategies is a promising area for future work. Decreasing or increasing the sample dimension of the embedding space ($m$) leads to lower performance, with the worst performance occurring at highest dimensional sampling. We postulate this performance drop occurs because the current distance measure in Equation (5) becomes less meaningful in higher-dimensional space (Park et al., 2024). Changing the score selection to use a uniformly random coverage sample $k$ decreases performance, which validates our design choice to focus coverage selection on embedding examples that occupy larger, lower-density areas. Removing redundancy score ($s^R$) decreases performance more substantially than any other ablative configuration,

Table 4: Comparison of Unlabeled and Labeled coreset selection methods on **ImageNet**. Full dataset training on the ResNet32 model training achieves 73.54% accuracy. Despite using unlabeled data, BlindCS has the best 90% prune rate performance. A results plot is provided in the Appendix.

| Method | Coreset Selection Requirements | 70% | 80% | 90% | Mean / Rel. Rand. |
|---|---|---|---|---|---|
| **BlindCS** | Unlabeled Data | **64.43** | **61.31** | **53.99** | **59.91** +0.72 |
| Random | Unlabeled Data | 64.19 | 60.76 | 52.63 | 59.19 +0.00 |
| TDDS CVPR 2024 | Full Training on Labeled Data | **64.69** | **62.56** | 53.91 | **60.39** +1.19 |
| Forgetting ICLR 2019 | Full Training on Labeled Data | 64.29 | 62.01 | 52.14 | 59.48 +0.29 |
| Moderate ICLR 2023 | Full Training on Labeled Data | 64.04 | 61.35 | 52.45 | 59.28 +0.09 |
| Entropy ICLR 2020 | Full Training on Labeled Data | 62.34 | 56.80 | 43.39 | 54.18 -5.02 |

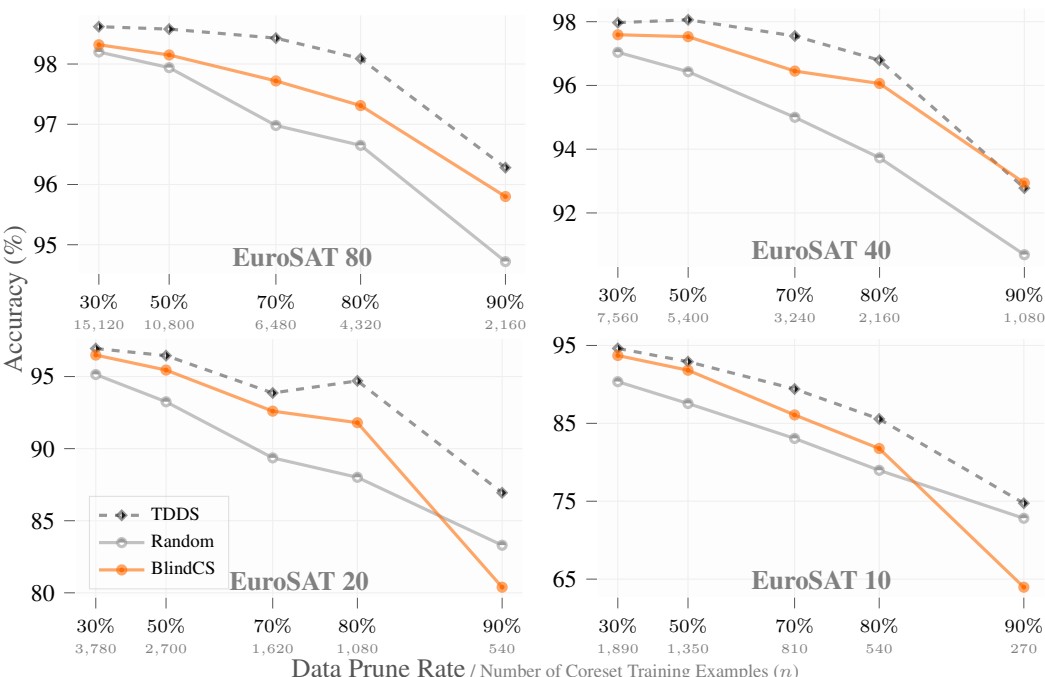

Figure 3: Comparison of Unlabeled (solid lines) and Labeled (dashed) coreset selection methods on 80/20, 40/60, 20/80, and 10/90 training/validation splits of **EuroSAT**. Dashed line indicates labeled coreset selection with training-based pruning. $x$-axis is in log scale for Number of Coreset Training Examples. BlindCS and TDDS prune selections outperform the mean accuracy of Random on all EuroSAT splits. A results table is provided in the Appendix.

which validates our design choice to penalize nearest neighbors in the embedding space to reduce redundancy. Finally, removing the score loss weight $w$ from model training decreases performance.

We plot the runtime and accuracy performance of BlindCS over a wide range of score iterations in Figure 4. The largest accuracy increase occurs when the coverage and redundancy score iterations ($T$) increase from 100 to 1,000, at which point, with the redundancy score reaching 1,000 neighbors per iteration, the score likely reaches most of the 50,000 CIFAR100 candidate training examples. Notably, the standard BlindCS configuration ($T = 1M$) runtime takes less than 400 s on a standard laptop, which, in addition to being able to select coresets for unlabeled data, makes BlindCS a computationally efficient alternative to label- and training-based coreset selection methods.

# 7    CONCLUSION

We motivate, formulate, and develop a method for unlabeled coreset selection, which enables data-*and* label-efficient deep learning relative to prior label-based coreset selection methods. Furthermore, unlike current SOTA methods, our approach requires no training on the dataset being con-

Table 5: Comparison of **BlindCS ablations** on CIFAR 100 with ResNet18. Accuracy is mean across 30%, 50%, 70%, 80%, and 90% prune rates over five repeat trials. "ResNet18, CLIP" is a concatenated embedding space that uses both off-the-shelf models.

| Ablation | Off-the-shelf Embedding Space Model | Sampling Distribution | Embedding Sample Dimension | Use Full Score | Mean CIFAR100 Accuracy |
|---|---|---|---|---|---|
| Full Method (**BlindCS**) | ResNet18, CLIP | Triangular | 2 | Yes | **65.77** |
| Embedding Space ($Z$) | **ResNet18** | Triangular | 2 | Yes | 65.52 |
| | **CLIP** | Triangular | 2 | Yes | 64.84 |
| Sampling Distribution (s) | ResNet18, CLIP | **Gaussian** | 2 | Yes | 65.75 |
| | ResNet18, CLIP | **Uniform** | 2 | Yes | 64.84 |
| Number of Embedding | ResNet18, CLIP | Triangular | **1** | Yes | 64.59 |
| Sample Dimensions ($m$) | ResNet18, CLIP | Triangular | **3** | Yes | 65.10 |
| | ResNet18, CLIP | Triangular | **10** | Yes | 63.27 |
| | ResNet18, CLIP | Triangular | **100** | Yes | 62.20 |
| Random Coverage Sample ($k$) | **NA** | **NA** | **NA** | **No** | 64.74 |
| No Redundancy Score ($s^{R}$) | ResNet18, CLIP | Triangular | 2 | **No** | 61.71 |
| No Score Loss Weight ($w$) | ResNet18, CLIP | Triangular | 2 | **No** | 63.53 |

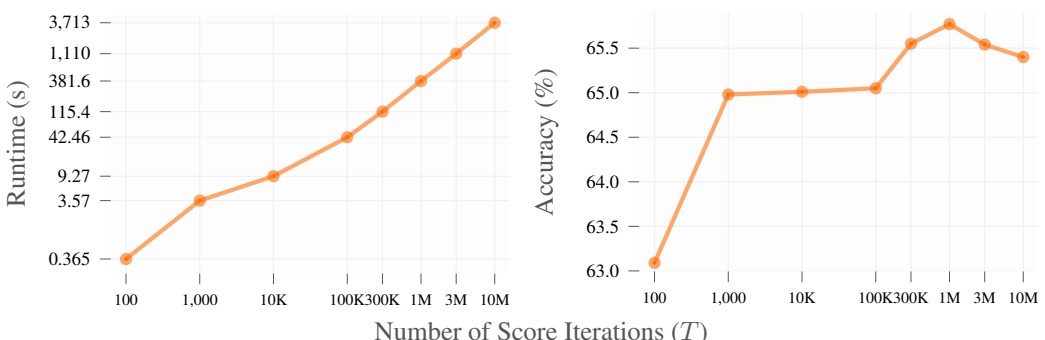

Figure 4: Comparison of number of score iterations vs. **runtime** (left) and accuracy (right) on CIFAR100 with ResNet18. Accuracy is mean across 30%, 50%, 70%, 80%, and 90% prune rates over five repeat trials. The accuracy peaks at 1 M iterations then converges on a slightly lower accuracy. Runtime experiments measure coreset selection times using a M3 Max-equipped laptop.

sidered for selection, which also reduces computation costs. We evaluate our method against the state-of-the-art using eight baselines across four datasets, ranging from initial datasets of over a million images all the way down to pruned coresets of 270 training images. In these experiments, our method outperforms all others save one, which requires full ground truth labels and model training on the initial dataset prior to coreset selection. However, our method alone does not use labels or dataset training, making it more efficient for coreset selection at the scale of current deep learning research. From these results, our method sets a new state-of-the-art for coreset selection.

In future work, to further improve performance on very small datasets, we will develop a sampling scheme that automatically determines the number of samples and nearest neighbors for redundancy scoring. In addition to the coverage and redundancy scores in this paper, we postulate that there are many more unlabeled features that can quantify coreset value for individual candidate examples. Furthermore, since there is no domain-specific limitation to our method, we will explore how it and the general coreset selection problem are applicable in other domains like point cloud and natural language and other problems like object detection and segmentation.

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

# A  APPENDIX

## A.1  REPRODUCIBILITY STATEMENT

We provide detailed experimental settings in Sections 5-6. We generate all BlindCS experimental results from a single attempt of five consecutive trials with the exception of ImageNet, which is from a single attempt of one trial. We will make our code publicly available with the final paper.

## A.2  ADDITIONAL TABLES & FIGURES

To supplement the evaluation in Section 6, we provide additional Figures and Tables. We plot coreset selection results for CIFAR10 and CIFAR100 in Figure 5, which demonstrates coreset selection for two medium-sized datasets. We plot coreset selection results for ImageNet in Figure 6, which demonstrates coreset selection at a large scale. We provide coreset selection results for all EuroSAT dataset splits in Table 6, which demonstrates coreset selection for the three leading methods at a much smaller scale.

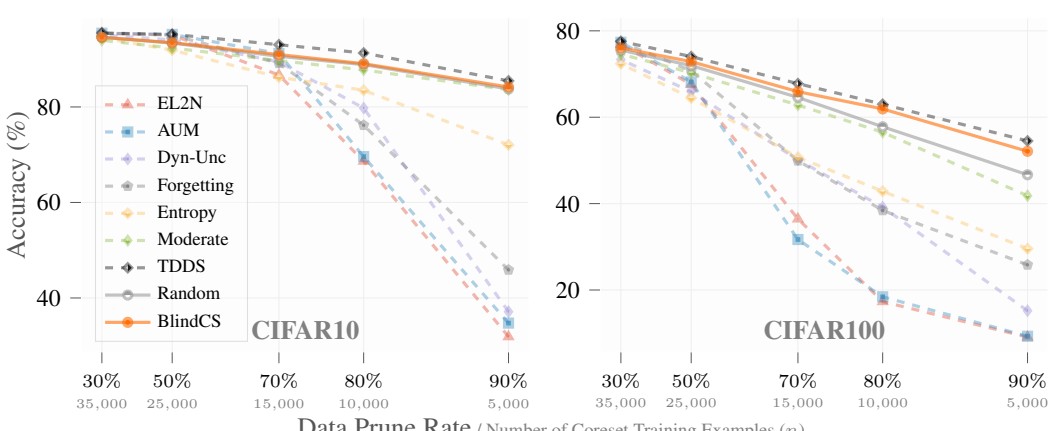

Figure 5: Comparison of Unlabeled (solid lines) and Labeled (dashed lines) coreset selection methods on **CIFAR10** and **CIFAR100**. Dashed line indicates labeled coreset selection with training-based pruning. $x$-axis is in log scale for Number of Coreset Training Examples. Notably, BlindCS and TDDS are the only methods outperforming Random, with the largest relative performance gaps between methods occurring at high prune rates.

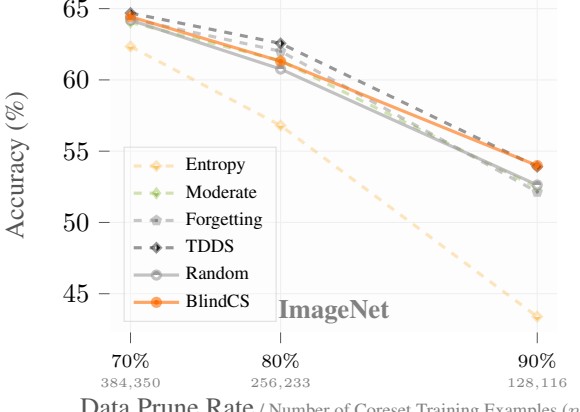

Figure 6: Comparison of Unlabeled (solid lines) and Labeled (dashed lines) coreset selection methods on **ImageNet**. Dashed line indicates labeled coreset selection with training-based pruning. $x$-axis is in log scale for Number of Coreset Training Examples. BlindCS achieves best 90% prune rate performance without using label- or training-based prune selection.

Table 6: Comparison of Unlabeled and Labeled coreset selection methods on different sized splits of **EuroSAT**. Full dataset training on the ResNet18 model achieves 98.59% (EuroSAT 80), 98.20% (EuroSAT 40), 98.59% (EuroSAT 20), and 93.64% (EuroSAT 10) accuracy. "Rel. Rand." is Mean accuracy across all prune rates relative to Random. "EuroSAT All" is Mean accuracy for all EuroSAT splits. BlindCS and TDDS prune selections outperform the mean accuracy of Random on all EuroSAT splits. Notably, the EuroSAT 10 90% prune rate coreset has only 270 training examples.

| Prune Method | Coreset Selection Requirements | Data Prune Rate / Number of Examples | | | | | Mean Rel. Rand. |
|---|---|---|---|---|---|---|---|
| | | 30% | 50% | 70% | 80% | 90% | |
| **EuroSAT All** | | | | | | | |
| BlindCS | Unlabeled Data | 96.53 | 95.74 | 93.21 | 91.74 | 83.27 | **92.10** +1.14 |
| Random | Unlabeled Data | 94.56 | 92.91 | 89.80 | 87.88 | 83.61 | 90.96 +0.00 |
| TDDS CVPR 2024 | Full Training on Labeled Data | 96.93 | 96.35 | 94.55 | 93.56 | 87.80 | **93.96** + 3.01 |
| **EuroSAT 80** (21,600) | | 15,120 | 10,800 | 6,480 | 4,320 | 2,160 | |
| BlindCS | Unlabeled Data | 98.32 ±0.08 | 98.15 ±0.13 | 97.72 ±0.13 | 97.31 ±0.17 | 95.80 ±0.18 | 97.46 +0.56 |
| Random | Unlabeled Data | 98.20 ±0.11 | 97.94 ±0.10 | 96.98 ±0.17 | 96.65 ±0.29 | 94.72 ±0.49 | 96.90 +0.00 |
| TDDS CVPR 2024 | Full Training on Labeled Data | 98.62 ±0.05 | 98.58 ±0.11 | 98.43 ±0.03 | 98.09 ±0.10 | 96.28 ±0.11 | 98.00 +1.10 |
| **EuroSAT 40** (10,800) | | 7,560 | 5,400 | 3,240 | 2,160 | 1,080 | |
| BlindCS | Unlabeled Data | 97.59 ±0.05 | 97.53 ±0.16 | 96.45 ±0.12 | 96.06 ±0.19 | 92.94 ±0.55 | 96.11 +1.54 |
| Random | Unlabeled Data | 97.04 ±0.07 | 96.43 ±0.37 | 95.00 ±0.67 | 93.73 ±0.58 | 90.69 ±0.53 | 94.58 +0.00 |
| TDDS CVPR 2024 | Full Training on Labeled Data | 97.97 ±0.09 | 98.06 ±0.06 | 97.55 ±0.08 | 96.79 ±0.16 | 92.78 ±0.27 | 96.63 +2.05 |
| **EuroSAT 20** (5,400) | | 3,780 | 2,700 | 1,620 | 1,080 | 540 | |
| BlindCS | Unlabeled Data | 96.49 ±0.16 | 95.45 ±0.22 | 92.60 ±0.29 | 91.80 ±0.70 | 80.39 ±3.91 | 91.35 +1.53 |
| Random | Unlabeled Data | 95.14 ±0.32 | 93.25 ±0.59 | 89.36 ±0.54 | 88.01 ±0.22 | 83.30 ±0.73 | 89.81 +0.00 |
| TDDS CVPR 2024 | Full Training on Labeled Data | 96.94 ±0.10 | 96.45 ±0.07 | 93.86 ±0.56 | 94.70 ±0.35 | 86.94 ±0.55 | 93.78 +3.97 |
| **EuroSAT 10** (2,700) | | 1,890 | 1,350 | 810 | 540 | 270 | |
| BlindCS | Unlabeled Data | 93.71 ±0.23 | 91.82 ±0.23 | 86.08 ±1.16 | 81.77 ±2.68 | 63.96 ±2.76 | 83.47 +0.92 |
| Random | Unlabeled Data | 90.35 ±0.64 | 87.55 ±0.67 | 83.06 ±1.61 | 78.97 ±1.88 | 72.81 ±2.25 | 82.55 +0.00 |
| TDDS CVPR 2024 | Full Training on Labeled Data | 94.62 ±0.09 | 92.92 ±0.33 | 89.41 ±0.52 | 85.56 ±0.67 | 74.74 ±2.02 | 87.45 +4.90 |