# OpenReview forum: "Blind Coreset Selection: Efficient Pruning for Unlabeled Data"
_ICLR.cc/2025/Conference — ICLR 2025 Conference Withdrawn Submission_

### Official Review · Reviewer_fEYP · 2024-10-23

**Soundness:** 3
**Presentation:** 3
**Contribution:** 2
**Rating:** 5
**Confidence:** 5

**Summary:**

This paper describes a technique for coreset selection from unlabeled data. The proposed method consists of three main steps. First, feature embeddings of the unlabeled data are extracted using different pre-trained models. In the second step, the feature distribution of unlabeled data is triangulated, and dimensionality reduction is applied to lower the feature dimensions. Finally, the third step involves selecting samples by eliminating redundant features after dimensionality reduction. The experimental comparisons are presented on four datasets.

**Strengths:**

1. The topic of how to solve the coreset selection problem for unlabeled data is both interesting and novel.
2. The presentation is clear and easy to understand.

**Weaknesses:**

1. In Table 1, it is questionable that BlindCS are not trained on the data. To be clear, BlindCS is not trained on the target dataset, but instead uses already trained pre-trained models (possibly by training on larger datasets), which is another way of using the training data. In addition, you need to emphasize that other methods do not use pre-trained models.
2. The experimental setup is unreasonable. BlindCS introduces the pre-trained models of resnet18 and CLIP, leading to an unfair comparison with other methods. The other coreset selection methods compared do not use pretrained models by default. For example,  assuming that the resnet18 pre-trained model can be used, other methods can be fine-tuned on resnet18 directly using the selected coreset.  However, BlindCS must use additional pre-trained models to extract features (e.g. resnet18, CLIP, etc.). Finally, The results of BlindCS has a great relationship with the pre-trained models, and the method has certain defects.
3. Formula (9) selects the nearest sample to the center of each subspace to form the coreset, so how to ensure the coverage rate of the selected sample under different pruning rates.
4. The ablation experiment is in doubt. There is concern about the choice of the parameter ${m}$, and ablation experiments show that the method is sensitive to the parameter ${m}$. Since BlindCS does not introduce labels, the selection can only be made on different dimensions after dimension reduction, but not on different classes. So how to ensure that the coreset does not have long tail problems.

**Questions:**

1. See the weakness section. Maybe elaborate more on your opinions about point 1-4.
2. In particular, BlindCS introduces pre-trained models, while previous baselines do not use pre-trained models. If pre-trained models can be used, existing coreset selection methods can still solve the unlabeled data problem, such as K-Center greedy.

---

> ### Author Response · Authors · 2024-11-13
>
> Thank you for your review.
>
> Regarding (1-2.) pre-trained models and experimental setup, great suggestion to specify which baselines methods use pre-trained models. See related comment 1. to Reviewer o99d: “The training of ResNet18 and CLIP has already taken place, with or without our paper. Furthermore, this training only takes place once, whereas the existing methods in the literature require subsequent training every time coreset selection is applied to a new dataset. Notably, we use CLIP embeddings from a constant prompt across all experiments with no specific textual information. We will add these details to the appendix.”
>
> Regarding (3.) coverage with different prune settings, the samples with the greatest coverage score (minus redundancy) will be selected regardless of specific prune rate (L76-L77, L315-317). Comparison of experimental results to 8 baselines indicate that this approach outperforms prior art across several dataset scales and prune rates (L343-349, L357-368, L370-418).
>
> Regarding (4.) selection of different classes, to clarify, as L177-203 explains, we use strictly unlabeled data for coreset selection. Specifically, our coreset selection method uses the unlabeled embedding space (L72-77, Section 5.1). To be clear, class and label information are unknown during the coreset selection process, but all dataset examples and classes are represented in the unlabeled embedding space (L197-215). Generally speaking, dataset examples corresponding to different classes will map to different regions of the embedding space, which we maximize coverage for (L276-281). Regarding $m$, the ablative experiments lead us to use $m=2$ across all other experiments (L281-285, L497-502). Would you please clarify what you mean by "the selection can only be made on different dimensions after dimension reduction, but not on difference classes?"
>
> Regarding (Q2.) pre-trained models, see related comment 1. to Ro99d. Notably, cluster-based methods like Herding (L149-152) are outperformed by cited work and similarly cluster-based baseline Moderate (Xia et al., 2023, ICLR), which our method outperforms across multiple datasets and prune settings (L365-367, L394, L441).

---

### Official Review · Reviewer_91AF · 2024-10-29

**Soundness:** 2
**Presentation:** 2
**Contribution:** 2
**Rating:** 3
**Confidence:** 4

**Summary:**

This paper presents a new method for unsupervised core-set selection by taking the coverage and redundancy of the embedding space.

**Strengths:**

1. The motivation of the paper is valid. Selecting a small subset of data for labeling can reduce costs, while many existing methods make an unrealistic assumption that training data is already labeled.
2. The proposed method is simple and easy to implement, which does not involve model training.
3. Experiments on several datasets show the advantage of the proposed method against previous methods.

**Weaknesses:**

1. The method proposed in this paper is highly heuristic, making it difficult to verify the optimality of core set selection. For example, the sampling strategy in Equation (3) and the feature random selection strategy in Equation (4) lack clear underlying principles, and their effectiveness has not been thoroughly analyzed in the experiments. Additionally, the principles and advantages of Equation (7) are not discussed. Therefore, the method in this paper employs a series of heuristic strategies to improve the efficiency of sample selection, but it does not provide an in-depth analysis of the superiority of the approach itself, nor does it offer theoretical performance guarantees.

2. Self-supervised learning baselines for core-set selection should be included in the experiments. As the primary goal of the paper is to reduce labeling costs, self-supervised learning offers a fair baseline for comparison since it eliminates the need for labeled data.

3. On the ImageNet dataset, the improvement of BlindCS over Random is minimal. Furthermore, for experiments involving randomly selected samples, the process should be repeated multiple times to report the average values and variance.

4. From Table 5, it is evident that Triangular and Gaussian sampling methods perform comparably. Additional results are needed to demonstrate the advantages and justify the effectiveness of Triangular sampling.

5. Hyperparameter analysis for $\alpha$ and $\beta$ is missing and should be included in the experiments to provide further insights.

6. In Equation (1), the expression $\frac{1 - n}{N}$ should be corrected to $1 - \frac{n}{N}$.

7. The paper uses $s$ to denote both the importance score and random sample, which may lead to confusion.

8.  Section 4 is too brief and can be integrated into Section 3 or Section 5 to improve the paper’s structure and flow.

**Questions:**

1. It is interesting to see if the proposed method can generalize to more pre-trained models in addition to ResNet18 and CLIP-L/14.

---

> ### Author Response · Authors · 2024-11-13
>
> Thank you for your review. Please see our detailed responses and experiment updates below.
>
> Regarding (1.) the optimality of coreset selection, as stated in the paper, finding the optimal coreset is NP-hard, which is why coreset selection is an active area of research (L145-156). Equations (3)-(4) are principally grounded in Monte Carlo sampling techniques to estimate the 1,280-dimensional embedding space coverage contribution of up to >1M candidate examples for coreset selection (L241-266, L234, L330). See comment 2. to Reviewer o99d for more discussion on Equation (3). Equation (4) increases sampling efficiency by reducing individual sample dimensionality while still covering all dimensions over the complete process of coreset selection (L281-284). Notably, Equation (7) is ablated in Section 6.3 (see L503 in Table 5). In absence of theoretical performance guarantees, we validate our method through exhaustive experiments (see comment to 3. in Ro99d), outperforming all prior work save TDDS with less computation (L234-L238) and uniquely without any dataset-specific labels or training (L483-485, L527-533).
>
> Comment 2. to Ro99d: “Regarding the advantages of Triangular distributions and their ability to cover long tails in real-world scenarios, we are in agreement. In terms of analysis and discussion, we included plots of several distributions in Figure 2 with corresponding discussion in L216-230 and L245-245 and ablative studies for Gaussian and Uniform distributions in L486-505 and L421-426. Notably, further exploring sampling strategies is already called out as an area of future work (L426).”
>
> Comment to 3. to Ro99d: “Regarding downstream tasks and theoretical explanation, current coreset selection experiments are evaluated using downstream model training across four datasets spanning over 1M images and 1,000 classes down to 2,700 images and 10 classes (L324-337, L370-418). To our knowledge, this is a greater span of downstream model training experiments in terms of dataset size and complexity than any existing coreset selection work. Theoretical explanations of the approach are provided in the Introduction (L54-78, L89-99) and Sections 4-5 (L177-323).”
>
> Regarding (2.) self-supervised learning, fantastic suggestion to use a self-supervised baseline to establish a result additionally without _pre-trained_ labels! See details in response to Q1 below.
>
> Regarding (3.) ImageNet results, following the precedent set by recent cited work in ICLR and CVPR, due to the large computational cost for the ImageNet training across settings, we train each model for one time. We will add this detail to the appendix. Regarding performance relative to random, our method and TDDS are the only methods amongst these recent related works to outperform random's average performance across all prune settings (L405-409, L432-442). The fact that it is not a statistically significant result relative to TDDS is irrelevant, the important thing is that we are _unlabeled_ and perform within error of TDDS, and are the only methods to outperform random.
>
> Regarding (4.) sampling distributions, see comment 2. to Ro99d.
>
> Regarding (5.) hyperparameter analysis, we performed no hyperparameter tuning for $\alpha$ and $\beta$ and changes to $\alpha$ are already called out as an area of future work (L415-417). If the reviewer insists, we will add additional ablative studies to Table 5 following completion of your suggestion for self-supervised experiments.
>
> Regarding (6.) prune rate, corrected in the revised manuscript. Great catch, thank you!
>
> Regarding (7.) nomenclature selection, these are not the same variable. Following the suggested ICLR nomenclature protocol, the importance score is $\vs$ and the random sample is $\rvs$. Nonetheless, we now see from your comment how this could cause confusion, and will update the importance score to use a different symbol (your specific recommendation is welcome).
>
> Regarding (8.) Section 4 organization, great suggestion! We will combine Sections 3 & 4 in the revised manuscript.
>
> Regarding (Q1.) expansion of embedding space models, fantastic suggestion! We are currently running additional repeat trials experiments with a self-supervised dinov2-vitb14 and pre-trained ResNet50 backbone accordingly. Thus far, performance is comparable to the existing results, and we will add these updates to the Ablation Study in Section 6.3.

---

### Official Review · Reviewer_o99d · 2024-11-04

**Soundness:** 2
**Presentation:** 3
**Contribution:** 2
**Rating:** 3
**Confidence:** 2

**Summary:**

The paper studies the problem of coreset selection for unlabeled data. The paper develops an unlabeled coreset selection method, Blind Coreset Selection (BlindCS), that jointly considers overall data coverage on a distribution as well as the relative importance of each example based on redundancy. The experiments demonstrate the performance of the proposed method.

**Strengths:**

1. The idea of the article is reasonable and the presentation of the problem is fine.

2. The paper proposes a new method for solving the problem of coreset selection for unlabeled data, which achieves comparable results to the supervised methods.

**Weaknesses:**

1. In the proposed method, to obtain the embedding space, already trained models (ResNet18 and CLIP in the paper) are required, and the overhead of training these models should be taken into account to compare with the supervised methods. In addition, the inputs to these models may actually contain more information such as textual information, which may lead to unfair evaluation.

2. The advantage of Triangular distributions may actually be due to a priori, such as modeling of the long tails, which is actually valid for some specific real-world scenarios. Although Triangular distributions are beneficial in covering the long tail of the distribution, actually change the distribution relative to the original distribution, as in the two examples shown in Fig. 2, therefore more detailed analysis and discussion are needed to demonstrate the advantages of such sampling.

3. The construction of a coreset generally depends on the particular downstream tasks. The coreset constructed in the paper is not closely linked to the downstream tasks, and lacks theoretical explanation, which weaken the reliability of the proposed method.

Some typos: line 229: ample->sample.
Eq. 10: s+min(s)->s-min(s)?

**Questions:**

In line 169, the prune rate (1-n)/N can be negative, right?

---

> ### Author Response · Authors · 2024-11-13
>
> Thank you for your review. We would appreciate the opportunity to accommodate your suggestions, but we need more specificity.
>
> Regarding (1.) the use of pre-trained foundational models, the training of ResNet18 and CLIP has already taken place, with or without our paper. Furthermore, this training only takes place once, whereas the existing methods in the literature require subsequent training every time coreset selection is applied to a new dataset. Notably, we use CLIP embeddings from a constant prompt across all experiments with no specific textual information. We will add these details to the appendix.
>
> Regarding (2.) the advantages of Triangular distributions and their ability to cover long tails in real-world scenarios, we are in agreement. In terms of analysis and discussion, we included plots of several distributions in Figure 2 with corresponding discussion in L216-230 and L245-245 and ablative studies for Gaussian and Uniform distributions in L486-505 and L421-426. Notably, further exploring sampling strategies is already called out as an area of future work (L426). Is there a new specific experiment you are requesting for the revised manuscript?
>
> Regarding (3.) downstream tasks and theoretical explanation, current coreset selection experiments are evaluated using downstream model training across four datasets spanning over 1M images and 1,000 classes down to 2,700 images and 10 classes (L324-337, L370-418). To our knowledge, this is a greater span of downstream model training experiments in terms of dataset size and complexity than any existing coreset selection work. Theoretical explanations of the approach are provided in the Introduction (L54-78, L89-99) and Sections 4-5 (L177-323). What specific downstream tasks or theoretical explanations are missing?
>
> L169 is 1-n/N in the revised manuscript. Great catch, thank you!

---

### Official Review · Reviewer_yXaZ · 2024-11-04

**Soundness:** 2
**Presentation:** 2
**Contribution:** 2
**Rating:** 3
**Confidence:** 5

**Summary:**

The paper proposed a pruning-based unlabeled data learning for the Coreset algorithm.

**Strengths:**

The proposal is interesting as it shows a new direction for unlabeled data learning for the Coreset algorithm.

**Weaknesses:**

The major weakness of the paper is that the paper lacks experiments.  The experimental section is very shallow. as only CIFER 10, 100 and imagenet are used. There exist many datasets in the literature that should be used, such as MSCOCO, VOC etc. More experiments on varying segmentation tasks, NLP, retrieval, tasks etc should be conducted to prove the proposal's effectiveness.

A comparison with many related works in the literature is also missing.

**Questions:**

What is the effectiveness of the algorithm on other task rather than classification.

---

> ### Author Response · Authors · 2024-11-13
>
> Thank you for your review.
>
> Regarding coreset selection for classification, we agree that our new method of unlabeled coreset selection is applicable to several domains (L537-539). Notably, we are following the precedent of classification experiments established by cited work in ICLR, NeurIPS, and CVPR, and our current experiments comparing against 8 baselines (L357-368) already put the current manuscript at the page limit.
>
> Would you please be more specific and provide examples of related coreset selection work that simultaneously study classification, segmentation, NLP, and the other domain areas you mentioned as missing?

---

### Author Response · Authors · 2024-11-13
**General Remarks**

We thank the editor and four reviewers for reviewing the paper. The reviewers commented that “the motivation of the paper is valid. Selecting a small subset of data for labeling can reduce costs, while many existing methods make an unrealistic assumption that training data is already labeled” (Reviewer 91AF);  “the paper proposes a new method for solving the problem of coreset selection for unlabeled data, which achieves comparable results to the supervised methods” (Ro99d); “how to solve coreset selection problem for unlabeled data is both interesting and novel” and “the presentation is clear and easy to understand” (RfEYP). We thank the reviewers for their many positive comments.

The paper’s stated contributions include: 1) motivating and formalizing the problem of unlabeled coreset selection to uniquely reduce data- _and_ label-based costs for deep learning at scale; 2) developing our Blind Coreset Selection method, which is computationally efficient and enables broader application relative to prior methods; and 3) evaluating our method against the state-of-the-art label- and training-based coreset selection methods with eight baselines on four different datasets spanning three orders of magnitude for scale, outperforming all methods save one despite our method being the only one to operate without labels or training. All three contributions are entirely undisputed.

This paper initiates a new methodology, so we appreciate helpful reviewer suggestions for peripheral improvements and clarifications that will improve the paper. Below, we address all reviewer comments in turn. We will add all new discussion to the revised manuscript.

---

### Note · Authors · 2024-11-15

I have read and agree with the venue's withdrawal policy on behalf of myself and my co-authors.